# Gold@Silica Nanoparticles Functionalized with Oligonucleotides: A Prominent Tool for the Detection of the Methylated Reprimo Gene in Gastric Cancer by Dynamic Light Scattering

**DOI:** 10.3390/nano9091333

**Published:** 2019-09-18

**Authors:** María José Marchant, Leda Guzmán, Alejandro H. Corvalán, Marcelo J. Kogan

**Affiliations:** 1Laboratorio de Química Biológica, Instituto de Química, Pontificia Universidad Católica de Valparaíso, 2373223 Valparaíso, Chile; marchant.mariajose@gmail.com (M.J.M.);; 2Departamento de Hematología y Oncología, Facultad de Medicina, Pontificia Universidad Católica de Chile, 8330032 Santiago, Chile; 3Advanced Center for Chronic Diseases (ACCDiS), Pontificia Universidad Católica de Chile, 8330034 Santiago, Chile; 4Departamento de Química Farmacológica y Toxicológica, Facultad de Ciencias Químicas y Farmacéuticas, Universidad de Chile, 8380494 Independencia, Santiago, Chile

**Keywords:** gold nanoparticles, methylated *RPRM* DNA, dynamic light scattering, gastric cancer, noninvasive detection

## Abstract

Reprimo (*RPRM*) is a tumor suppressor gene involved in the development of gastric cancer. Hypermethylation of the *RPRM* promoter region has been found in tumor tissue and plasma samples from patients with gastric cancer. These findings suggest that circulating methylated DNA of *RPRM* could be a candidate for a noninvasive detection of gastric cancer. We designed a nanosystem based on the functionalization of silica coated gold nanoparticles with oligonucleotides that recognize a specific DNA fragment of the *RPRM* promoter region. The functionality of the oligonucleotide on the surface of the nanoparticle was confirmed by polymerase chain reaction (PCR). The nanoparticles were incubated with a synthetic DNA fragment of methylated DNA of *RPRM* and changes in the size distribution after hybridization were evaluated by dynamic light scattering (DLS). A difference in the size distribution of nanoparticles hybridized with genomic DNA from the KATO III gastric cancer cell line was observed when was compared with DNA from the GES-1 normal cell line. These results showed that this nanosystem may be a useful tool for the specific and sensitive detection of methylated DNA of *RPRM* in patients at risk of developing gastric cancer.

## 1. Introduction

Gastric cancer (GC) is one of the most serious malignant tumors worldwide despite the global decrease in incidence over the last decades [1]. Despite recent progress in surgery and chemotherapy, the prognosis for gastric cancer is still unfavorable. [1]. Unfortunately, at early stage, GC develops without symptoms, so most tumors are not diagnosed until late stage, with low survival rates. Therefore, early detection at curable stages is urgently needed. To achieve this goal, noninvasive detection through a simple blood test is necessary [2]. Reprimo (*RPRM*) is a highly glycosylated protein that was initially discovered as a putative tumor-suppressor involved in the regulation of the p53-dependent G2 arrest of the cell cycle [3]. The presence of *RPRM* promoter methylation has been detected in tumor tissues and plasma from gastric cancer patients [4,5]. Therefore, the detection of circulating methylated DNA of *RPRM* in samples of plasma could be a candidate for a noninvasive detection strategy. However, at early stages of the disease (i.e., dysplasia), the detection rate for methylated DNA of *RPRM* is low [6]. To solve this problem, the development of technologies that allow for the detection of methylated DNA of *RPRM* circulating in the blood via GC samples obtained noninvasively is mandatory.

Nanotechnology has emerged as a tool that improves the sensitivity of various techniques used to analyze macromolecules, such as proteins and DNA in different biological samples (e.g., human blood serum, plasma, and tissue extracts) [7]. Gold nanoparticles (AuNPs) have been of great interest due to the fact of their large light absorption and scattering cross-section in the surface plasmon resonance wavelength regions. Another property of AuNPs is the surface plasmon resonance (SPR), which corresponds to the oscillation of electrons confined to the surface of a nanoparticle upon interaction with electromagnetic radiation [8]. This interaction with electromagnetic radiation is dependent on molecules, such as water, ions or DNA, that are on the surface of the nanoparticle [9]. Thus, the interaction of AuNPs functionalized with an oligonucleotide of methylated DNA of *RPRM* could have an effect on the light scattering, which is associated with the presence of a biomarker on surface of the nanoparticle. This property could be exploited to develop a nanosensor utilizing the dispersion of light as a tool for improving the detection of biomarkers [10,11].

Dynamic light scattering (DLS) is a technique widely used for studies of particle size and size distribution on the basis of the Brownian motion of particles, analyzing the temporal fluctuations by means of the photon scattering auto-correlation function [12]. The DLS produced by a nanoparticle depends on the molecules that are present on the surface, as well as the type and concentration of ions in the medium. These factors can modify the size of the nanoparticle, generating a population of nanoparticles with different aggregation states [12]. The interaction among AuNPs functionalized with DNA strands could form structures called hybrids, as is shown in the Scheme 1. These hybrids show a characteristic size distribution, with a shift to the right due to the presence of DNA strands on the surface, and its effect on the light scattering of the nanoparticles. Moreover, in the case of AuNPs, the SPR effect can strongly enhance the light scattering effect and, accordingly, increase the signal by improving the sensitivity of the designed nanosensor [13]. 

The aim of this work was to functionalize AuNPs with DNA strands complementary to a sequence of methylated DNA from the *RPRM* promotor region. After the hybridization process, the distribution sizes of the nanoparticles were analyzed by DLS. In the presence of methylated DNA from *RPRM*, the functionalized nanoparticles make hybrids, generating a variation in the nanoparticle’s aggregation state. It was possible to detect significant differences in the size distributions of these hybrids (methylated DNA of *RPRM* versus unspecific DNA). The obtained results in this study showed that is possible to detect a low number of molecules of methylated DNA from *RPRM*.

## 2. Materials and Methods 

### 2.1. Cell Lines

The human GC cell line (KATO III) employed in this study was obtained from the American Type Culture Collection (ATCC, Manassas, VA, USA). In addition, we employed the human gastric epithelial cell line (GES-1) as a control for healthy cells (kindly donated by Dawit Kidane-Mulat from the University of Texas–Austin).

### 2.2. Reagents

Chloroauric acid (HAuCl_4_), (3-aminopropyl) trimethoxysilane (APTMS) 97% *w*/*v*, sodium silicate solution, and tetraethyl orthosilicate (TEOS) reagent grade 98% *w*/*v* were purchased from Sigma–Aldrich (St. Louis, MO, USA). Sodium citrate, dimethyl sulfoxide (DMSO), ammonia, and ethanol were purchased from Merck (Darmstadt, Germany). All chemicals from commercial sources were of analytical grade or the highest purity available. Milli-Q water (resistance of ~18.2 MΩ, filtered through a 0.2 µm filter) was utilized for the preparation of working solutions. 

### 2.3. Oligonucleotides

The oligonucleotides were obtained from Integrated DNA Technologies (Coralville, IA, USA). The recognition zone of a DNA sequence and its specificity were previously described [14]. The modified oligonucleotide sequence was: 5’-TACCTAAAACCGAATTCATCG/iSp18/3AmMO. The iSp18 modifier was an 18-atom hexa-ethylene glycol spacer and the 3AmMO was an amine group modifier attached at the 3’ end to enable conjugation with the modified nanoparticles. The oligonucleotide sequence used for the polymerase chain reaction (PCR) assay was: 5’- GCGAGTGAGCGTTTAGTTC-3’.

### 2.4. Synthesis of AuNPs

The synthesis of 40 nm AuNPs was performed via the reduction of HAuCl_4_ using citric acid, as previously described by Turkevich et al. [15]. Citrate-capped AuNPs were characterized by UV-Vis spectroscopy Epoch ELISA reader (ELx800, BioTek, Winooski, VT, USA), DLS (Zetasizer Nano ZS, Malvern, UK), and scanning transmission electron microscope (STEM). The electron micrographs were analyzed with ImageJ software (version 1.8, NIH, Maryland, USA) over an average of at least 100 particles. 

### 2.5. Synthesis of Core–Shell Au@SiO2–COOH 

In a colloidal solution, 40 nm AuNPs were coated with silica by the Stöber method [14]. Briefly, 50 mL of AuNPs were reacted with 0.25 mL of 1 mM APTMS for 15 min at 25 °C under vigorous magnetic stirring. The solution of active silica at low pH was prepared from a 0.54 wt % sodium silicate solution at pH 10 using a cation exchange resin (Amberlite^®^ IR120 hydrogen form, Sigma–Aldrich). Then, 2 mL of active silica was added to 50 mL of the solution (1:4 water/ethanol) containing the silica shell particles and maintained under vigorous magnetic stirring for 24 h. Then, 0.15 mL of TEOS and 1 mL of ammonia were added, and the solution was maintained for another 12 h under mild magnetic stirring. To modify the surface of Au@SiO_2_ with carboxyl groups (COOH), 50 mL of particles with silica were reacted with 0.25 mL of 1 mM APTMS for 15 min. Then, 500 mg of succinic anhydride were dissolved in 1 mL of DMSO for 24 h at 25 °C under magnetic stirring. Finally, the nanoparticles were subsequently washed 3 times with Milli-Q water to remove excess reagents and were resuspended in nuclease-free water, followed by the UV-Vis spectroscopy, DLS, and STEM characterizations.

### 2.6. Functionalization of Au@SiO_2_–COOH with Oligonucleotides 

First, 30 µL of Au@SiO2–COOH (OD λ_max_ = 50) were mixed with 10 µL of freshly prepared 20 mg/mL EDC (1-ethyl-3-(3-dimethylaminopropyl) carbodiimide) solution. The reaction was incubated for 30 min at room temperature under stirring. Then, the nanoparticles were recollected and resuspended in 0.25 mL of an oligonucleotide solution (0.04 µM in nuclease free water). This solution was incubated for 2 h at room temperature with mixing. Subsequently, the colloids were recollected by centrifugation three times, resuspended in 1 mL of nuclease-free water (0.25 nM final concentration of nanoparticles) and were characterized by UV-Vis spectroscopy, DLS, and STEM.

### 2.7. DNA Samples

A synthetic DNA fragment (Au@SiO2–COOH-oligo) of 112 bp for methylated DNA of (IDT, DNA Technologies, Coralville, IA, USA) was used to assess the successful of functionalization. Three dilutions containing 50, 500, and 5000 copies of methylated DNA of were utilized for hybridization assays. Also, an in vitro approach was performed using two cell lines (GES-1 and KATO III). Genomic DNA from cell lines were extracted using an EZNA^®^ Tissue DNA Kit (Omega Biotek, Norcross, GA, USA) according to the manufacturer´s instructions. First, 1 μg of total DNA extracted was modified with the EZ DNA Methylation Kit™ Gold, according to the manufacturing instructions. All samples were aliquoted and stored at −20 °C until be used.

### 2.8. DNA_met-RPRM_ Captured by Hybridization and PCR Assay

To evaluate the accessibility and recognition capacity of the functionalized oligonucleotide on the surface of the nanoparticles, a capture assay and PCR were performed. Twenty microliters of Au@SiO2–COOH-oligo was incubated with 20 ng of methylated DNA of *RPRM* (previously denatured at 95 °C for 5 min) in PBS1X for 30 min at 37 °C and 300 rpm. Then, the mix was washed 3 times with nuclease-free water, and the obtained pellet was used to perform the PCR reaction. The PCR was carried out using 0.25 μM primers described in Section 2.3, 4 μL buffer 5X (KAPA HiFi with MgCl_2_), 0.2 mM of 10 mM dNTPs, Au@SiO2–COOH-oligo hybridized with a methylated DNA of *RPRM* synthetic fragment, and 0.2 μL KAPA HiFi polymerase. A reaction without DNA was used as a negative control. The thermal cycling program consisted of 94 °C for 5 min, followed by 40 cycles of denaturation at 94 °C for 40 s, annealing at 58 °C for 30 s, and extension at 72 °C for 45 s, and then a final extension at 72 °C for 7 min. The PCR products were analyzed by 2% agarose gel electrophoresis and staining with ethidium bromide.

### 2.9. Hybridization Assay

The hybridization assay was performed by incubating the functionalized nanoparticles with the different DNA samples (previously denatured at 95 °C for 5 min) in PBS1X for 30 min at 37 °C under vigorous stirring. Subsequently, these samples were washed once by centrifugation and resuspended in 80 μL nuclease-free grade water. To determine the size distributions, measurements were made via DLS using the Malvern Zetasizer Nano ZS (Malvern Instruments) equipment with disposable 301 polycarbonate capillary cell cuvettes at 25 °C. Each reaction was performed at least in triplicate, and each measurement was performed in quintuplicate for the statistical analysis of the data.

### 2.10. Statistical Analysis 

To identify significant differences, we compared the control reaction (without DNA) or DNA control (unmethylated DNA of *RPRM*) to each of the other conditions using a one-way ANOVA for non-parametric data in GraphPad Prism 7.0a Software (GraphPad Software, San Diego, CA, USA).

## 3. Results and Discussion 

### 3.1. Characterization of Gold Nanoparticles

Using the citrate reduction method, we obtained AuNPs and characterized them by UV-Vis, DLS, and STEM, as indicated in Figure 1. The UV-Vis absorption spectrum recorded from AuNPs showed the characteristic SPR band of AuNPs centered at 528 nm (Figure 1a). Figure 1b shows a representative STEM image of AuNPs, which were obtained as a monodisperse colloid of spherical nanoparticles. Subsequently, the AuNPs were chemically modified and functionalized to obtain the nanosensor. Table 1 summarizes the characterization of the different AuNPs modified. The data were obtained for three independent functionalizations, with five measures for each reaction.

The DLS analysis showed an increase in size after each modification step (silica coating and COOH modification) and functionalization with the oligonucleotide. In addition, the zeta potential values showed changes in the surface charge due to the effect of the dielectric constant on the gold surface, through which changes in the light scattering properties of AuNP were generated after the modifications. The STEM image of Au@SiO_2_ showed a dark center, which corresponded to a gold core, and a gray halo, which corresponded to the silica coating (Figure 2a). The UV-Vis absorption spectra of the different modified nanoparticles showed a redshift that can be attributed to the increase in the size of the nanoparticle after the silica coating and functionalization with the oligonucleotide (Figure 2b and Table 1).

Moreover, the stability assay evaluating Au@SiO2–COOH-oligo with different concentrations of NaCl solution was performed to evaluate the presence of oligonucleotides on the surface of nanoparticles by UV-Vis spectroscopy. The results showed that the Au@SiO2–COOH-oligo colloid was stable in solution up to 1 M NaCl (Appendix A). This stability can be attributed to the interaction of the phosphate groups present in the oligonucleotide molecule with the sodium ions, decreasing the electrostatic repulsion between nanoparticles while maintaining the characteristic SPR band, with a maximum absorption at 530–540 nm. However, when the concentration of sodium ions surpasses the equilibrium (Appendix A, 2 M NaCl), a displacement in the SPR of the aggregation of the nanoparticles is observed due to the absence of electrostatic repulsion between nanoparticles.

### 3.2. Capture Assay and Functionality of Au@SiO2–COOH-Oligo 

According to the previous results (Table 1), we observed that the nanoparticles were functionalized successfully demonstrating the presence of the oligonucleotide on the surface. Moreover, using a PCR assay, we obtained amplicons, indicating the functionality of the oligonucleotide on the surface of the nanoparticles (Figure 3). The PCR products were visualized in a 2% agarose gel stained with ethidium bromide. The black arrow indicates the fragment corresponding to the methylated DNA of the promoter region of the *RPRM* gene of 112 bp. In the Figure 3, line 5 corresponds to the methylated DNA of *RPRM* reacting with free oligonucleotides in solution, where unspecific bands were observed. On the other hand, in line 3, in the presence of an oligonucleotide coupled to the nanoparticle (Au@SiO2–COOH-Oligo), only one band was detected. These PCR results confirm that the immobilized oligonucleotides on the gold nanoparticle surfaces maintain their accessibility to perform PCR contributing also to reducing the presence of unspecific bands. This effect was also described previously for the detection of the nitrogen fixation gene in *Azospirillum* bacteria and for other DNAs [16,17,18]. Taken together, here we demonstrated that an oligonucleotide anchored to the Au@SiO2–COOH is accessible for further recognition by methylated DNA of *RPRM*.

### 3.3. Hybridization Assay of Au@SiO2–COOH-Oligo with a Synthetic Fragment of RPRM-DNA Analyzed by DLS

As was shown, Au@SiO2–COOH-oligo is a stable and functional nanoparticle, which can be used for the detection of methylated DNA of *RPRM*. Dynamic light scattering allows the detection of changes in the distribution size attributable to the nanoparticle interaction with the methylated DNA of *RPRM* biomarker. Previous reports have shown that the interaction between functionalized nanoparticles and different biomarkers such as proteins and DNA produce hybrids that can be analyzed by DLS [19,20]. Therefore, by analyzing the size distribution of nanoparticles, it is possible to find individually dispersed hybrids, dimers, trimers, and large aggregations formed due to the hybridization between Au@SiO2–COOH-oligo and methylated DNA of *RPRM*. However, the presence of different proportions of nanoparticle aggregates produces a shielding effect, whereby the light scattering of the more abundant population suppresses light scattering produced by the less abundant population, thus, decreasing the detection by DLS [21]. Consequently, the change in distribution sizes of the Au@SiO2–COOH-oligo nanoparticles before and after the addition of DNA was analyzed.

Initially, a hybridization assay of Au@SiO2–COOH-oligo in the presence of three different amounts (50, 500, and 5000 copies) of the synthetic DNA fragment of methylated DNA of *RPRM* was evaluated, and changes in the size distribution of the majority population for each reaction performed were analyzed. The results obtained are shown in Figure 4 and are summarized in Table 2, showing an average (size and percentage of NPs) of all the experiments performed for each condition, corresponding to the size of the highest peak in the DLS data. Significant differences were observed between the control reaction without DNA and the serial dilutions of the synthetic DNA fragment of methylated DNA of *RPRM*. The DLS results showed three populations of nanoparticles after hybridization with methylated DNA (Appendix A). The population of nanoparticles corresponding to the negative control (without DNA, reaction 0) showed that from the total number of nanoparticles analyzed by DLS, 98% exhibited a size of 151 nm. This was the major peak observed and which corresponded to non-hybridized nanoparticles. However, the reactions with the 50, 500, and 5000 copies showed a 10% decrease in the average, in respect to the negative control. With the size distribution profiles, a small population of nanoparticles of greater size (~4000 nm) can be observed, which would correspond to aggregates induced by the interaction of Au@SiO2–COOH-oligo with methylated DNA (Appendix A). The electrostatic interactions of DNA strands with AuNPs reduce the interparticle repulsion and induce the formation of AuNP aggregates, which could correspond to the nanoparticle population of the largest size [19]. Also, a peak at 20 nm was observed which can be attributed to molecules of methylated DNA of *RPRM* that did not hybridize with the Au@SiO2–COOH-oligo nanoparticles in the reaction mix (Appendix A). These results demonstrate that Au@SiO2–COOH-oligo can be used for the detection of methylated DNA of *RPRM* by DLS.

### 3.4. Hybridization Assay of Au@SiO2–COOH-Oligo with DNA from Cell Lines Analyzed by DLS

Another important aspect of a nanosensor is the specificity; that is, the ability to discriminate between the target molecule and others with similar characteristics. To evaluate this property, the hybridization assay was performed in the presence of DNA obtained from the GES-1 and KATO III cell lines. The GES-1 is a normal gastric epithelium cell line, so it is an in vitro model that can represent a healthy patient. On the other hand, KATO III is a GC cell line, for which previous studies have reported the presence of methylated DNA of *RPRM* [22]; therefore, the KATO III cell line represents a cancer patient. The results obtained are shown in Figure 5 and are summarized in Table 2. The percentage of nanoparticles was compared with the control reaction (without DNA). A decrease in the percentage of nanoparticles in the reaction with DNA from KATO III was observed. The size distribution by DLS revealed a change in the size of nanoparticles corresponding to the most abundant populations for both GES-1 and KATO III cell lines. In the case of the GES-1 cell line, 96% of the nanoparticle population had an average size of 249 nm. The increase in the average size can be attributed to the nonspecific interaction of genomic DNA with the oligonucleotide attached to the surface of the AuNPs. However, this interaction was unable to induce aggregation of the nanoparticles and generate a significant change in the size distribution, which is similar to previously reported results [23].

Differences between the size of the distribution profiles of the hybrids generated using a synthetic DNA fragment of *RPRM* methylated and the hybrids generated with genomic DNA from GES-1 and KATO III cell lines were observed (Appendix A). These results can be attributed to the size of DNA used in the hybridization assay as—a synthetic DNA fragment (174 pb) of single strand of methylated DNA of *RPRM*—with respect to hybrids generated with a genomic DNA modified by sodium bisulfite, resulting in DNA fragments above 1000 bp. Therefore, the hybrids formed by the interaction with genomic DNA had a larger size compared to those observed with the synthetic DNA fragment of methylated DNA of *RPRM*. In the case of the hybridization assay with genomic DNA from KATO III, 68% of the population of nanoparticles showed a size of 240 nm, similar to the size observed with genomic DNA from GES-1 cell line. This decrease in the population is similar to that observed in the hybridization assay with the synthetic DNA fragment of methylated DNA of *RPRM*. Additionally, the size distribution profile shows a population of nanoparticles with a size about 1000 nm, which may correspond to hybrids formed by the interaction of Au@SiO2–COOH-oligo with methylated DNA of *RPRM* from the KATO III cell line (Appendix A). Consequently, these results demonstrate the specificity of the designed nanosensor. The nanoparticles were able to recognize a specific sequence of methylated DNA of *RPRM* in a reaction containing genomic DNA. 

## 4. Conclusions

Noninvasive detection of GC could reduce mortality and improve the overall survival rate of patients. Novel technologies that allow for the detection of sequences of circulating methylated DNA of *RPRM* are urgently needed. For this reason, our work focused on the use of DLS, as an analytical technique, to detect a specific nanosensor of silica-coated gold nanoparticles functionalized with oligonucleotides specifically designed to detect methylated DNA of *RPRM*. The AuNPs were successfully synthesized by citrate reduction and subsequently coated with silica and functionalized with the oligonucleotide. Characterization by UV-Vis, DLS, and STEM indicated that the obtained nanomaterial was stable and homogenous. Moreover, the PCR assay showed that the oligonucleotide was functional, which is essential for the detection of a DNA target. The hybridization assay with the synthetic DNA fragment of *RPRM* methylated revealed that the nanoparticles interacted with the DNA forming hybrids. These structures then formed aggregates, reducing the percentage of free nanoparticles in the reaction. Similar results with genomic DNA from the GC cell line demonstrated the specificity of the nanosystem to discriminate between sequence of methylated DNA of *RPRM* and unmethylated genomic DNA. Therefore, our results demonstrate that the developed nanosensor based on silica-coated nanoparticles coupled to DLS could be a useful tool for the ultrasensitive detection of methylated DNA of *RPRM* in clinical samples from patients at risk to develop GC.

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
