# Peer review of "Gold@Silica Nanoparticles Functionalized with Oligonucleotides: A Prominent Tool for the Detection of the Methylated Reprimo Gene in Gastric Cancer by Dynamic Light Scattering"

_nanomaterials, 2019, doi:10.3390/nano9091333_

Round 1

Reviewer 1 Report

The authors have developed a system to detect a gastric cancer biomarker based on the functionalization of gold nanoparticles with a specific oligonucleotide with recognizes a region methylated RPRM-DNA and that can be monitored by dynamic light scattering (DLS) to detection. Clearly, the changes in the DLS are associated with the aggregation state of the nanoparticles. That’s why I think it could be interesting to discuss more in depth the changes in the Z-potential of the nanoparticles before and after the recognition event. Also, the authors ascribe the response of the system to bigger aggregates and relate it with the percentage of nanoparticles retaining the initial size. In the KATO case, only 30% of nanoparticles were transformed, which would mean the formation of 3-4 aggregates? Maybe the authors could represent the obtained data in number of nanoparticles to have a more clear vision. In any case, further discussion of the low rate of the process could be very useful for future investigations.

Other minor items:

The abbreviation of gastric cancer GC is used in the abstract without previous identification.

Please review chemical formulas subscripts

Please review the degree symbol within all the manuscript

Author Response

RESPONSES TO COMMENTS FROM REVIEWER 1

Sincerely thanks to reviewer 1 for his valuable comments. Our responses (in BLUE) to all the author comments (in BLACK color) are listed below:

Reviewer Comments:

Comment 1: That’s why I think it could be interesting to discuss more in depth the changes in the Z-potential of the nanoparticles before and after the recognition event.

Response: The Z-potential was determined only for characterization of gold nanoparticles before and after the functionalization process. This parameter was not determined in the recognition event because the volume of the hybridization reaction was low (100 mL).  However, it would be interesting to evaluate changes in this parameter for future investigations.

Comment 2: In the KATO case, only 30% of nanoparticles were transformed, which would mean the formation of 3-4 aggregates? Maybe the authors could represent the obtained data in number of nanoparticles to have a more clear vision. In any case, further discussion of the low rate of the process could be very useful for future investigations.

Response: The aggregation process of the nanoparticles is particularly difficult to establish, so the number of aggregates was impossible to be determined. Additionally, the data obtained from the number of nanoparticles was not reproducible between replicas of the same experiment. Therefore, the intensity was a parameter more reproducible and representative of the hybridization process and consequently this parameter was chosen for the analysis.

Comment 3: The abbreviation of gastric cancer GC is used in the abstract without previous identification.

Response: The abbreviation has been modified.

Comment 4: Please review chemical formulas subscripts 

Response: The abbreviations has been revised.

Comment 5: Please review the degree symbol within all the manuscript.

Response: Has been revised.

Reviewer 2 Report

This manuscript presents the synthesis and characterization of gold@silica core-shell nanoparticles that are functionalized with methylated Reprimo (RPRM) DNA for the sensitive detection of methylated RERM-DNA for potential early detection of gastric cancer (GC). Characterization techniques include UV-Vis extinction spectroscopy, dynamic light scattering (DLS), zeta potential measurements, scanning transmission electron microscopy (STEM), and polymerase chain reaction (PCR) assay. The nanoparticles show the ability for sensitive detection of methylated RPRM-DNA using DLS, both in clean prepared solutions and from two different cell lines, using suitable control experiments. While this work is very interesting and promising, I have many questions and comments that I think need to be addressed.  

(1) My main line of questions centers on the presentation and analysis of the DLS data. Figures 4 and 5 show “Percent Nanoparticles” of the largest DLS peak. But in the Supplementary Materials, the entire DLS size distributions are shown, which contain much more information. Why are only the “Percent Nanoparticles” shown in the manuscript? In Table 2, the “Size” and Percentage of NPs” are listed from the DLS measurements. Is that the percentage at that corresponding size peak in the DLS data? Is that the size of that maximum peak or is it the overall size taking into account all different peaks and the relative intensities? This isn’t clear. Was this the analysis done automatically by the DLS instrument or were the different peaks fit, for example using Gaussian functions for cases of overlapping peaks, to determine the overall percentages of the main peaks.

(2) In the DLS measurements in Figure S2, there are several peaks in B, C, and D. What do these peaks correspond to? It’s strange that the largest peak seems to get smaller in size after the RPRM-DNA interaction. Why is this? And why is there are new peak at much lower size at around 20 nm? This needs to be discussed. In addition, there is also a much larger peak at around 5000 nm. What is this from? Is this from aggregation? Perhaps the UV-vis spectrum can also be shown to help determine this, after the nanoparticles interact with the RPRM-DNA to help determine if aggregation is occurring.

(3) A key emphasis of this study is that the functionalized nanoparticles can have better sensitivity for the RPRM-DNA than is currently available. However, looking at Figure S2, it seems there is not sufficient data to determine the sensitivity of this technique. DLS data at the lowest RPRM-DNA concentration in B seems to be the same as at the highest RPRM-DNA concentration in D. Perhaps lower concentrations, below 50 copies should be studied to determine the level of sensitivity of the measurement.

(4) The DLS results shown in Figure S3 look very different than in Figure S2. Why is this? GES-1 shows a broad shoulder while KATO III shows a pronounced second peak. Why is this? And neither of the new peaks in Fig S3 B and C correspond to the new peaks shown in Fig S2. If the data can’t be accurately understood, I would have to question the overall interpretations or the reliability of the technique.

(5) From the STEM analysis, what is the standard deviation of the size of the gold nanoparticles? The histogram in Fig 1A never reaches zero so it’s difficult to tell. Also, the spectrum in Figure 1A has a y-axis of “Absorbance (R.U.)”. What are R.U. units? Why not just use regular extinction units? Also, for gold nanoparticles, the spectrum has significant components from both absorption and scattering, so I think “extinction” is more accurate than “absorption”. Also, in this spectrum, the baseline at 700 nm to 800 nm is pretty high. For clean gold nanoparticles, this should correspond to theory, either from Mie theory or another model. In this case, it seems these nanoparticles may have a lot of aggregation to have such a large non-zero baseline. Also, on line 121, the OD is given as “50 a.u.”. Again, please use standard extinction units, not arbitrary units, to help other researchers understand your work better.

(6) On lines 64-66, the description of DLS does not seem to be accurate, especially for the instrument being used. My understanding is that the technique uses time-domain scattering correlation functions, not Doppler shift analysis, to determine the particle size distributions. Can the authors please check this and update this section accordingly with references, if necessary?

(7)  Finally, there are many instances of incorrect English, especially in the Abstract and the Introduction. Additionally, “GC” is sometimes “CG”. Many acronyms such as PCR and STEM are not defined either in their first use or at all.

Overall, I think this work represents a very interesting study. However, I feel these questions above need to be answered and addressed before I can recommend publication in this journal.

Author Response

RESPONSES TO COMMENTS FROM REVIEWER 2

Sincerely thanks to reviewer 2 for his valuable comments. Our responses (in Blue Color) to all the author comments (in BLACK color) are listed below:

Comment 1: My main line of questions centers on the presentation and analysis of the DLS data. Figures 4 and 5 show “Percent Nanoparticles” of the largest DLS peak. But in the Supplementary Materials, the entire DLS size distributions are shown, which contain much more information. Why are only the “Percent Nanoparticles” shown in the manuscript?

Response: Two smaller peaks are observed in figure S2 and S3. DLS analyzes of these peaks showed a variable average size of nanoparticles after hybridization assay, being impossible to establish correlations between the DNA concentration and nanoparticles size. However, the highest peak (non-hybridized nanoparticles) has a comparable average size between experiments performed with the same DNA sample, hence was possible to analyze statistically the data.

Comment 2: In Table 2, the “Size” and Percentage of NPs” are listed from the DLS measurements. Is that the percentage at that corresponding size peak in the DLS data? Is that the size of that maximum peak or is it the overall size taking into account all different peaks and the relative intensities? This isn’t clear.

Response: Thank you for the comment. We agree with referee. In order to avoid misunderstanding, the next sentence has been incorporated in the section 3.3: “The Size and Percentage of NPs are an average of all experiment performed for each condition and corresponds to the size of the highest peak in the DLS data.”

Comment 3: Was this the analysis done automatically by the DLS instrument or were the different peaks fit, for example using Gaussian functions for cases of overlapping peaks, to determine the overall percentages of the main peaks.

Response: The analysis was performed automatically by the DLS instrument.

Comment 4: In the DLS measurements in Figure S2, there are several peaks in B, C, and D. What do these peaks correspond to? It’s strange that the largest peak seems to get smaller in size after the RPRM-DNA interaction. Why is this? And why is there are new peak at much lower size at around 20 nm? This needs to be discussed. In addition, there is also a much larger peak at around 5000 nm. What is this from? Is this from aggregation? Perhaps the UV-vis spectrum can also be shown to help determine this, after the nanoparticles interact with the RPRM-DNA to help determine if aggregation is occurring.

Response: Thank you for the comment. The question about the larger peak was answered in the section 3.3. We analyzed the hybridization assay by UV-Vis spectroscopy, however, it was not possible observed significative changes the UV-vis spectrum.

Comment 5: A key emphasis of this study is that the functionalized nanoparticles can have better sensitivity for the RPRM-DNA than is currently available. However, looking at Figure S2, it seems there is not sufficient data to determine the sensitivity of this technique. DLS data at the lowest RPRM-DNA concentration in B seems to be the same as at the highest RPRM-DNA concentration in D. Perhaps lower concentrations, below 50 copies should be studied to determine the level of sensitivity of the measurement.

Response: Thank you for the comment. There are no significant differences between 50, 500, and 5000 copies of methylated RPRM DNA using DLS. These results show an on-off response of nanosystem. It would be interesting to evaluate less DNA to determine the sensitivity of the proposed method.

Comment 6: The DLS results shown in Figure S3 look very different than in Figure S2. Why is this? GES-1 shows a broad shoulder while KATO III shows a pronounced second peak. Why is this? And neither of the new peaks in Fig S3 B and C correspond to the new peaks shown in Fig S2. If the data can’t be accurately understood, I would have to question the overall interpretations or the reliability of the technique.

Response: Thank you for the comment. The differences observed in figures S2 and S3 are discussed in section 3.4. However, new comment was incorporated into this section to clarify the results shown.

Comment 7: From the STEM analysis, what is the standard deviation of the size of the gold nanoparticles? The histogram in Fig 1A never reaches zero so it’s difficult to tell.

Response: Thank you for the comment. In order to avoid misunderstanding, the next sentence has been incorporated in the description of figure 1: with an average size of 46  7 nm.

Comment 8: Also, the spectrum in Figure 1A has a y-axis of “Absorbance (R.U.)”. What are R.U. units? Why not just use regular extinction units? Also, for gold nanoparticles, the spectrum has significant components from both absorption and scattering, so I think “extinction” is more accurate than “absorption”.

Response: Thank you for the comment. The legends of the y-axes have been modified.

Comment 9: Also, in this spectrum, the baseline at 700 nm to 800 nm is pretty high. For clean gold nanoparticles, this should correspond to theory, either from Mie theory or another model. In this case, it seems these nanoparticles may have a lot of aggregation to have such a large non-zero baseline.

Response: The baseline at 700 to 800 nm is pretty high. However, that’s not a problem as the nanoparticles are centrifuged and washed before of silica coating and oligonucleotide functionalization. Subsequently, this was confirmed analyzing the same region of the UV-vis spectrum of modified and functionalized nanoparticles (Figure 2a).

Comment 10: Also, on line 121, the OD is given as “50 a.u.”. Again, please use standard extinction units, not arbitrary units, to help other researchers understand your work better.

Response: Thank you for the comment. The suggestion has been accepted.

Comment 11: On lines 64-66, the description of DLS does not seem to be accurate, especially for the instrument being used. My understanding is that the technique uses time-domain scattering correlation functions, not Doppler shift analysis, to determine the particle size distributions. Can the authors please check this and update this section accordingly with references, if necessary?

Response: Thank you for the very relevant comment. Indeed the described technique based in Doppler shift analysis is for the determination of the zeta potential. The description for DLS has been corrected in the introduction section.

Comment 12: Finally, there are many instances of incorrect English, especially in the Abstract and the Introduction. Additionally, “GC” is sometimes “CG”. Many acronyms such as PCR and STEM are not defined either in their first use or at all.

Response: The text has been revised.

Reviewer 3 Report

The authors are presenting a work on the detection of methylated DNA by gold-silica nanoparticles functionalized with oligonucleotides. The work could be interesting, but most of the claims related to the methylated DNA detection have not been demonstrated by data and results. Please, find specifici comments here below: 

-Abstract: please, write GC in full;

-Abstract: the sentence "...to recognizes a region methylated RPRM-DNA in biological samples, using the dynamic light scattering (DLS) to detection" contains several english errors and should be corrected;

-Abstract: the sentence "...the nanosystem development was able to discriminate.." should be corrected (the nanosystem was able to discriminate and not its development);

-Introduction: the sentence "Therefore, to detect GC at curable stages is urgent, to this propose is necessary to develop noninvasive detection approaches as “screening technique” through a simple blood test for the early detection of CG would make a difference in the survival outcomes for people with this disease."  is too long and not clear in this form, please correct;

-section 2.2: please, clarify the meaning of "iSp18/3AmMO" on the oligonucleotide;

-section 2.2: plesae, specify how the probe (oligonucleotide) was chosen, were its sequence is located in the methylated RPRM-DNA complete sequence, which is its specificity on the basis of BLAST analysis or similar;

-section 2.7: please, specify the sequence of the primers for the PCR;

-section 2.7 and section 2.8: methylated RPRM-DNA is a double stranded DNA. Is this double stranded DNA denatured before hybridisation with the probe on to the nanoparticles? Usually, in hybridisation assays, the target double stranded DNA is opened to obtain a single strand which can hybridise with the probe. Which method has been used in this case? If the DNA was not treated, how the authors can explain its hybridisation with the probe?

-Table 1: the oligonucleotide immobilized onto the nanoparticles is negatively charged like all the sequences of DNA. This immobilization should make the zeta potential more negative. How the authors explain the more positive zeta potential in presence of an oligonucleotide which is negatively charged?

-section 3.2: the claim "This result confirms that the oligonucleotide was coupled to the surface of the nanoparticle and presented the capability to recognize methylated RPRM-DNA." is not demonstrated. The authors should demonstrate that the methylated RPRM-DNA is effectively captured mùby the immobilized probe and not just adsorbed onto the nanoparticles by adding similar measurements (PCR) after the interaction of only Au@SiO2-COOH without the oligonucleotide with the target DNA;

-section 3.2: the claim "Therefore, Au@SiO2-COOH-Oligo enhances the hybridization efficiency of the oligonucleotide and the specificity of the amplification during PCR reaction..." is not demonstrated. The authors are just showinga gel electrophoresis of a PCR reaction which is not demonstrating the enhancement of the hybridisation efficiency and of the specificity;

-section 3.2: no data are presented to demonstrate the claim "These effects optimize the PCR reaction by i) limiting the formation of nonspecific products and ii) decreasing the amount of oligonucleotide required to achieve amplification";

-section 3.2: no data are presented to demostrate the claim "Additionally, the use of Au@SiO2-COOH-Oligo decreased the amount of oligonucleotide needed to amplify methylated RPRM-DNA by at least 125 times with respect to the reaction without nanoparticles. Consequently, the use of Au@SiO2-COOH improves the functionality and specificity of the oligonucleotide in the PCR reaction."

-section 3.3: plasma has not been used in the work, so the sentence "Plasma is a sample of great interest because it allows the study and/or detection of biomarkers for the diagnosis and monitoring of therapeutic treatments of various diseases in a noninvasive form."  should be deleted;

-section 3.3: "The presence of methylated RPRM-DNA in the bloodstream is a common event in GC; however, the levels in early stages are too low for reliable detection by conventional methods.": please clarify (in numbers) how much is the concentration of methylated RPRM-DNA in the bloodstream;

-section 3.3: "These results demonstrate that Au@SiO2-COOH-Oligo can be used for the ultrasensitive detection of methylated RPRM DNA by DLS.": please, clarify why the term "ultrasensitive" has been used. Where is this ultrasensitivity coming from? Which is the sensitivity of the proposed method and how this sensitivity is comparing with  other methods to detect methylated DNA? Ultrasensitivity should be demonstrated. If not, please avoid this term in the whole manuscript including the abstract;

-section 3.4: "The results obtained by the Au@SiO2-COOH-Oligo  nanoparticles confirm that this nanosensor development is suitable for to realize a specific and ultrasensitive detection of methylated RPRM-DNA in noninvasive samples as plasma.": plasma has not been tested. Please, delete all the sentences claiming the possibility of detecting methylated DNA in plasma which has not been demonstrated;

-conclusions: plasma and ultrasensitive shouldbe deleted; moreover, most of the claims have not been demonstrated.

Author Response

RESPONSES TO COMMENTS FROM REVIEWER 3

Sincerely thanks to reviewer 3 for his valuable comments. Our responses (in Blue color) to all the author comments (in BLACK color) are listed below:

Reviewer Comments:

Comment 1: Abstract: please, write GC in full

Response: The abbreviation has been modified.

Comment 2: Abstract: the sentence "...to recognizes a region methylated RPRM-DNA in biological samples, using the dynamic light scattering (DLS) to detection" contains several english errors and should be corrected.

Response: Thank you for the comment. The sentence has been amended.

Comment 3: Abstract: the sentence "...the nanosystem development was able to discriminate.." should be corrected (the nanosystem was able to discriminate and not its development).

Response: Thank you for the comment. The sentence has been amended.

Comment 4: Introduction: the sentence "Therefore, to detect GC at curable stages is urgent, to this propose is necessary to develop noninvasive detection approaches as “screening technique” through a simple blood test for the early detection of CG would make a difference in the survival outcomes for people with this disease." is too long and not clear in this form, please correct.

Response: Thank you for the comment. The sentence has been amended.

Comment 5: section 2.2: please, clarify the meaning of "iSp18/3AmMO" on the oligonucleotide.

Response: Thank you for the comment. The meaning has been specified in the section 2.3.

Comment 6: section 2.2: please, specify how the probe (oligonucleotide) was chosen, were its sequence is located in the methylated RPRM-DNA complete sequence, which is its specificity on the basis of BLAST analysis or similar.

Response: This information has been incorporated in the section 2.3 in the new version of the manuscript.

Comment 7: section 2.7: please, specify the sequence of the primers for the PCR.

Response: This information has been incorporated in the section 2.3 in the new version of the manuscript.

Comment 8: section 2.7 and section 2.8: methylated RPRM-DNA is a double stranded DNA. Is this double stranded DNA denatured before hybridization with the probe on to the nanoparticles? Usually, in hybridization assays, the target double stranded DNA is opened to obtain a single strand which can hybridize with the probe. Which method has been used in this case? If the DNA was not treated, how the authors can explain its hybridization with the probe?

Response: Thank you for the important comment. In order to obtain the single strand of methylated RPRM-DNA the sample was previously denatured at 95°C for 5 min. This is now described in sections 2.8 and 2.9 in the new version of the manuscript.

Comment 9: Table 1: the oligonucleotide immobilized onto the nanoparticles is negatively charged like all the sequences of DNA. This immobilization should make the zeta potential more negative. How the authors explain the more positive zeta potential in presence of an oligonucleotide which is negatively charged?

Response: Thank you for the important comment. The Au@SiO2-COOH-Oligo nanoparticles were analyzed in PBS 1X. The Na+ ions present in the solution stabilize the negative charges of the oligonucleotide to form the double layer of the colloid, so the zeta potential is more positive.
We specify in legend to table 1 the solvent that we used for each sample.

Comment 10: section 3.2: the claim "This result confirms that the oligonucleotide was coupled to the surface of the nanoparticle and presented the capability to recognize methylated RPRM-DNA." is not demonstrated. The authors should demonstrate that the methylated RPRM-DNA is effectively captured by the immobilized probe and not just adsorbed onto the nanoparticles by adding similar measurements (PCR) after the interaction of only Au@SiO2-COOH without the oligonucleotide with the target DNA

Response: We understand the claim. The aim of this experiment is only to demonstrate that the oligonucleotide is anchored to the nanoparticle and that this oligonucleotide is accessible for the hybridization with RPRM-DNA. In order to avoid misunderstanding, a new paragraph has been incorporated in section 3.2 in order to reinforce the concept that the oligonucleotide immobilized on the surface of the nanoparticle is functional.
…” The bands observed in Figure 3, line 5 correspond to the methylated RPRM-DNA reacting with free oligonucleotides in solution observing also an unspecific band, while in line 3 in the presence of oligonucleotide coupled to the nanoparticle (Au@SiO2-COOH-Oligo) was detected only one band. This result confirms that the oligonucleotide immobilized on the gold nanoparticle surface maintain the accessibility of the oligonucleotide for the recognition to perform PCR contributing also to reduce the presence of unspecific bands. This effect was also described previously for the detection of the nitrogen fixation gene in Azospirillum bacteria oligonucleotide anchored to the Au@SiO2-COOH is accessible for a further recognition by RPRM-DNA. These results allow us to demonstrate that the oligonucleotid anchored to the Au@SiO2-COOH is accessible for a further recognition by RPRM-DNA.”

Comment 11: section 3.2: the claim "Therefore, Au@SiO2-COOH-Oligo enhances the hybridization efficiency of the oligonucleotide and the specificity of the amplification during PCR reaction..." is not demonstrated. The authors are just showing a gel electrophoresis of a PCR reaction which is not demonstrating the enhancement of the hybridization efficiency and of the specificity.

Response: Thank you for the comment. Please see response to comment 11.

Comment 12: section 3.2: no data are presented to demonstrate the claim "These effects optimize the PCR reaction by i) limiting the formation of nonspecific products and ii) decreasing the amount of oligonucleotide required to achieve amplification"

Response: Thank you for the comment. Please see response to comment 11.

Comment 13: section 3.2: no data are presented to demonstrate the claim "Additionally, the use of Au@SiO2-COOH-Oligo decreased the amount of oligonucleotide needed to amplify methylated RPRM-DNA by at least 125 times with respect to the reaction without nanoparticles. Consequently, the use of Au@SiO2-COOH improves the functionality and specificity of the oligonucleotide in the PCR reaction"

Response: The suggestion has been considered and the sentence has been deleted.

Comment 14: section 3.3: plasma has not been used in the work, so the sentence "Plasma is a sample of great interest because it allows the study and/or detection of biomarkers for the diagnosis and monitoring of therapeutic treatments of various diseases in a noninvasive form." should be deleted.

Response: The sentence has been deleted.

Comment 15: section 3.3: "The presence of methylated RPRM-DNA in the bloodstream is a common event in GC; however, the levels in early stages are too low for reliable detection by conventional methods.": please clarify (in numbers) how much is the concentration of methylated RPRM-DNA in the bloodstream.

Response: The sentence has been deleted. However, the correlation between plasma/tumors pairs samples has been reported (Bernal et al., 2008), but the copy number in early stage of gastric cancer was not determined.

Comment 16: section 3.3: "These results demonstrate that Au@SiO2-COOH-Oligo can be used for the ultrasensitive detection of methylated RPRM DNA by DLS.": please, clarify why the term "ultrasensitive" has been used. Where is this ultrasensitivity coming from? Which is the sensitivity of the proposed method and how this sensitivity is comparing with other methods to detect methylated DNA? Ultrasensitivity should be demonstrated. If not, please avoid this term in the whole manuscript including the abstract.

Response: The term was avoided in the whole manuscript. In a future, we will study the sensibility of nanosensor design with methylated RPRM DNA in clinical samples.

Comment 17: section 3.4: "The results obtained by the Au@SiO2-COOH-Oligo nanoparticles confirm that this nanosensor development is suitable for to realize a specific and ultrasensitive detection of methylated RPRM-DNA in noninvasive samples as plasma.": plasma has not been tested. Please, delete all the sentences claiming the possibility of detecting methylated DNA in plasma which has not been demonstrated.

Response: The sentence has been corrected in the new version of manuscript.

Comment 18: conclusions: plasma and ultrasensitive should be deleted; moreover, most of the claims have not been demonstrated.

Response: The suggestion has been considered.

Round 2

Reviewer 1 Report

The manuscript can be published in its present form

Author Response

Reviewer #1

Comment: The manuscript can be published in its present form

Response: Thanks for your comments.

Reviewer 2 Report

Although not all of my questions and comments were adequately addressed, I believe the authors have substantially improved the paper such that I can now recommend this for publication. I would still like to see better sensitivity measurements (seeing what lower concentration of DNA is needed to see a change in the DLS results) and a better understanding of the different size distribution peaks in the DLS results, but perhaps these can be done in future studies.

There are still many minor English and grammar errors. After these minor English and grammar errors are corrected, I would recommend publication. I do not need to review this paper again.

Author Response

Reviewer #2

Comment 1: Although not all of my questions and comments were adequately addressed, I believe the authors have substantially improved the paper such that I can now recommend this for publication. I would still like to see better sensitivity measurements (seeing what lower concentration of DNA is needed to see a change in the DLS results) and a better understanding of the different size distribution peaks in the DLS results, but perhaps these can be done in future studies.

There are still many minor English and grammar errors. After these minor English and grammar errors are corrected, I would recommend publication. I do not need to review this paper again.

Response: Thanks for your comments and of course, we will include your suggestions in future studies. In this revised version, we have revised the English.

Reviewer 3 Report

The authors have clarified some points, but a couple of issues are still to be considered.

Please, find specific comments here below:

Abstract: in the last line of the abstract, the authors are still referring to analysis in plasma, and the possibility of using this method for this kind of analysis has not been demonstrated.

Line 214: To demonstrate that “the oligonucleotide immobilized on the gold nanoparticle surface maintain the accessibility”, the authors should include the results of the PCR performed with methylated RPRM-DNA incubated with the nanoparticles without the capture oligonucleotide. In this way they can demonstrate that the methylated DNA is actually captured onto the nanoparticles by the active probe and that the methylated DNA is not just adsorbed onto the nanoparticles.

Section 3.3.: also in this section, a control measurement with methylated RPRM-DNA and Au@SiO2-COOH (without oligo) should be included to demonstrate that the modifications in particle size distribution are actually due to the hybridization between the probe onto the nanoparticles and not only to the presence of DNA which is affecting the nanoparticle distribution and size, e.g. by adsorption onto the nanoparticles.

Author Response

Reviewer #3

Comment 1: Abstract: in the last line of the abstract, the authors are still referring to analysis in plasma, and the possibility of using this method for this kind of analysis has not been demonstrated.

Response: Thank you for the comment. In order to avoid misunderstanding, the sentence was modified.

Comment 2: Line 214: To demonstrate that “the oligonucleotide immobilized on the gold nanoparticle surface maintain the accessibility”, the authors should include the results of the PCR performed with methylated RPRM-DNA incubated with the nanoparticles without the capture oligonucleotide. In this way they can demonstrate that the methylated DNA is actually captured onto the nanoparticles by the active probe and that the methylated DNA is not just adsorbed onto the nanoparticles.

Response: Thank you for the comment. The PCR experiments had been performed also with methylated RPRM-DNA incubated with Au@SiO2-COOH. This result corresponds to the line 6 (Figure 1, not showed in Figure 3 of the manuscript). As can be seen in Figure 1, no bands were observed in the presence of the nanoparticles (lane 6) incubated with methylated DNA of RPRM. However, it is important to mention that in the hybridization conditions (in the presence of PBS 1X) the Au@SiO2-COOH exhibited some degree of aggregation (see answer to this referee comment number 3). For that, reason we did not included this result in the figure 3 of the manuscript..

Figure 1. Methylated DNA of RPRM capture by hybridization and the PCR reaction with Au@SiO2-COOH-Oligo. M: Molecular Weight 50 bp NEB®. bp: base pairs.

Comment 3: Section 3.3.: also in this section, a control measurement with methylated RPRM-DNA and Au@SiO2-COOH (without oligo) should be included to demonstrate that the modifications in particle size distribution are actually due to the hybridization between the probe onto the nanoparticles and not only to the presence of DNA which is affecting the nanoparticle distribution and size, e.g. by adsorption onto the nanoparticles.

Response: Thank you for your observation. This control is relevant, however we regret that is not possible to be performed due to the instability of Au@SiO2-COOH (without oligo) in PBS (conditions in which the hybridization has been performed). By UV-Vis, spectrophotometry was possible to visualize the aggregation effects in the presence of PBS for Au@SiO2-COOH (without oligo). Visible spectra of Au@SiO2-COOH nanoparticles in water (red line) and PBS 1X (black line) were shown in Figure 2. A spectral redshift of the nanoparticles in PBS 1X solution was observed, which suggests a partial aggregation due to the instability of the colloid. Additionally, the possibility of DNA adsorption on the surface of the X was discarded with the PCR assay (see the response to comment 2 to this referee).

Figure 2. UV-Vis spectra of the Au@SiO2-COOH nanoparticles in water and in PBS 1X solution.

Round 3

Reviewer 3 Report

The authors have considered the reviewer comments.